# Patterns of Myometrial Invasion in Endometrial Adenocarcinoma with Emphasizing on Microcystic, Elongated and Fragmented (MELF) Glands Pattern: A Narrative Review of the Literature

**DOI:** 10.3390/diagnostics11091707

**Published:** 2021-09-18

**Authors:** Svetlana Mateva, Margarita Nikolova, Angel Yordanov

**Affiliations:** 1Department of Pathoanatomy, Medical University Pleven, 5800 Pleven, Bulgaria; matevasvetlana@gmail.com; 2Department of Pathology Laboratory, Saint Marina University Hospital, 5800 Pleven, Bulgaria; mnikol@abv.bg; 3Department of Gynecologic Oncology, Medical University Pleven, 5800 Pleven, Bulgaria

**Keywords:** endometrial cancer, myometrial invasion, microcystic, elongated and fragmented glands, single cell invasion

## Abstract

Endometrioid endometrial adenocarcinoma (EEC) is the most common malignancy of the female genital tract. According to the 2009 FIGO staging system, the depth of myometrial invasion (MI), and tumor spread to adjacent organs or tissues are the staging criteria for endometrial carcinoma (EC). Therefore, assessment of the depth of MI is of great importance. There is a spectrum of morphological patterns of MI. Still, their number and features vary according to the scientific literature, with a certain overlap that creates difficulties and controversies in the precise assessment of MI depth. The purpose of this review is to present and discuss the most important and recent information about patterns of MI, focusing on the more aggressive and the elongated and fragmented glands (MELF) pattern in particular. Assessment of MI depth and correct staging of EC is possible only after the precise recognition of each MI pattern.

## 1. Introduction

Endometrial adenocarcinoma (EC) is the most frequent gynecological malignancy in developed countries with an overall good prognosis [1,2,3]. Many factors, such as age, histological type and grade, depth of myometrial invasion (MI), lymphovascular invasion (LVI), tumor size, and metastases in pelvic and lumbar-aortic lymph nodes, have a proven prognostic value [4,5]. EC is classified by Bokhman as two types. Type I consists of 80–90% of all EC, most of them well differentiated and estrogen-receptor positive. Type II are less frequent but represent aggressive tumors with poor prognosis, predominantly serous or clear cell morphology and not related to estrogenic stimulation [6].

Endometrioid endometrial carcinoma (EEC) is the most common histologic type of EC. The majority of the patients are classified as per Soslow et al. as low-grade tumors (G 1 and 2) [7], low-stage (IA-II, as defined by Sadowski et al.) [8]. These EEC have a favorable prognosis [5,9,10,11], but a small number of them may eventually develop unpredictable recurrence leading to an adverse prognosis [10,12,13]. According to risk stratification systems, low-grade and early-stage EECs are classified into low or intermediate-low risk groups. So, it is necessary to find new parameters to identify patients at a higher risk of relapse among early-stage diagnoses in order to treat and follow them properly without overtreating other patients who remain with an excellent prognosis. This is one of the main gaps that needs further improvement in the classification and staging of the disease and this issue needs to be addressed in the near future [10,14]. According to the 2009 FIGO staging system, the depth of MI and the tumor spread to adjacent organs or tissues are the current staging criteria for EC [15]. Zaino et al. reported that the 5-year survival rate of patients with tumors confined to the endometrium (FIGO stage IA) was 94%, versus 84% and 59% for tumors invading the middle and outer thirds of the myometrium, respectively [16]. Therefore, the correct assessment of the depth of the MI is of great importance. The existing spectrum of different morphological patterns of MI introduces difficulties in the precise assessment of the depth of MI. The MI pattern in low-grade, low-stage EEC is considered to be a potential predictive factor for tumor evolution [10,17,18]. Different patterns of MI have been described and more than one pattern may be found in the same tumor. The current recommendations for the prognosis assessment are based on the most aggressive pattern. In such cases, the current recommendations for the prognosis assessment are based on the most aggressive pattern [18,19]. The total number of all morphological patterns of MI varies according to different authors, with a certain overlap in some of the reports [5,18,19,20,21,22,23].

## 2. Aim

This narrative review aims to discuss the most important and recent information about MI patterns and to highlight the differences between them, with a focus on the more aggressive three types: microcystic, elongated, and fragmented glands (MELF), single-cell invasion (SCI) and diffusely infiltrative pattern. The assessment of the depth of MI with a subsequent correct staging of the EC is possible only after precise recognition of all patterns of MI, taking into consideration each of these patterns. Identification of more aggressive MI patterns would lead to selection of patients who need more aggressive treatment [17].

## 3. Discussion

The diffusely infiltrative pattern or single gland pattern is the most common morphological aspect observed at the invasive front in EC, observed in 47.8–89% [17,18,24,25]. Clement and Young characterize it as individually dispersed glands widely scattered throughout the myometrium [26]. Quick et al. define this pattern as invasion by a single or small group (three or less) of glands with irregular contours with or without a desmoplastic stromal response [18]. This pattern of invasion indicates poor prognosis as it is characterized with higher histological grade [17], LVI, and tumor recurrence [18,20]. It is associated with an increased expression of cancer stem cells (CSCs) markers—CD44 and CD133 and with a loss of estrogen receptors (ER), progesterone receptors (PR), and decreased E-cadherin. All these changes participate in the induction of the epithelial-mesenchymal transition (EMT) process, which we will be further discussed. Recent research has identified a particular type of stem cell, which is considered as responsible for the processes of invasion, metastasis, and the development of resistance to conventional therapy. Moreover, tumor milieu has reciprocal interactions with malignant cells and the stromal matrix is inducing CSC proliferation. The epithelial cells undergo EMT, followed by invasion and metastasis. Additional adaptive processes occur such as hormonal, chemotherapy, and radiotherapy resistance acquisition. Several markers have emerged as useful for the identification of CSCs—and the expression of CD44 and CD133 was closely associated with disease progression and poor prognosis [17]. The diffusely infiltrative pattern of MI was observed in many other types of malignancies [27].

The pushing/expansile, broad front/pattern is reported with variable frequency from 2.8% to 21% [18,25]. It is described as a large swath of neoplastic glands with well-defined margins that appear to push into the underlying myometrium with or without a desmoplastic response [14,19]. This pattern of MI is difficult to distinguish from the non-invasive EC. Some morphological criteria are strong indicators of pushing invasion: the presence of a linear front, as opposed to an undulating contour, and a desmoplastic response at the advancing edge of EC [20,24]. Actually, this issue always arises in practice: to correctly distinguish the invasion of the basal endometrium from the superficial myometrial invasion in EEC. However, it is considered that there is no difference of survival rates between superficial invasive EC and that which invades less than 1/2 of the myometrial thickness (90% versus 91%) [28]. This type of MI is associated with a good prognosis and low expression of CSC markers and EMT markers, as listed below [17]. Only one study has reported an association between this pattern of MI in low-grade, low-stage EEC and recurrence risk, which impacts the disease-free survival of the patients [22].

The third pattern of MI spread is the adenomyosis-like, which is reported with variable frequencies of 1.1% and 26.3%, respectively [17,25]. It represents groups of at least three malignant glands invading the myometrium in scattered islands [18] (Figure 1).

This pattern of MI should not be confused with adenomyosis involved by EEC and this is a challenge for pathologists because the latter cannot be considered when evaluating depth of myoinvasion (Figure 2).

The criteria suggesting adenomyosis-like MI pattern are: no residual endometrial stroma or glands around the tumor nests, irregular borders, desmoplastic stroma possibly with edema and inflammation, and the identification of similar foci of invasion within the adjacent myometrium [20,29]. This MI pattern is also associated with a good prognosis, reduced expression of CD44, CD33, and Nanog1, and preserved expression of E-cadherin, ER, and PR [17].

The adenoma malignum-like pattern is an exceedingly rare type of myometrial invasion with reported frequency of 0.2% to 1.33% [18,25]. It is composed of diffusely infiltrating regular, rounded, and widely spaced “naked” glands with a low cytological grade with no stromal or inflammatory response [17,18,19]. This deceptively bland morphology may be easily overlooked [20]. In EC, adenoma malignum pattern brings difficulties in assessing the tumor, grossly and microscopically. The diffuse, infiltrative spread of the glands, without an associated stromal reaction, can be misleading when appreciating the depth of MI [30]. Longacre et al. [19] proposed the use of the following histologic features to help diagnose the adenoma malignum-like pattern of the EC: The presence of focal cytologic atypia.Pseudostratification of glands.Focal loose, edematous stromal response (not desmoplasia).The presence of inflammatory cell infiltrate around problematic glands.Lymph vascular space invasion.

The adenoma malignum-like pattern can extend beyond the uterine body, frequently involving the cervix or, more rarely, the ovary [31] It is suggestive of a more aggressive behavior [18], despite the fact that some studies report that EC with this pattern of invasion does not have a worse evolution compared to those with a conventional tumor front [18,19]. EC with this pattern of MI has preserved the expression of ER, PR, CK7, and vimentin and loss of CK20, CEA, and p16 expression. This immunohistochemical (IHC) profile maybe useful in cases of cervical involvement in order to exclude a synchronous cervical adenocarcinoma, which on IHC is positive for CEA and p16 and is negative for vimentin and ER [31,32]. The adenoma malignum-like areas must also be distinguished from adenomyosis, deeply located endocervical glands, mesonephric remnants, cervical tubo-endometrioid metaplasia, cervical or ovarian endometriosis or cortical inclusion cysts of the ovary [31].

The phyllodes tumor-like pattern is characterized by nodular fasciitis-like stroma (vaguely nodular, myxoid with variable cellularity but without nuclear atypia, pleomorphism or mitotic activity and, collagen deposition) In this MI pattern, there are large cystic or elongated slit-like glands, lined by flattened cells with variable degrees of squamous differentiation. The large stromal nodules, along with the interconnected glandular spaces give resemblance to the phyllodes tumor of the breast. Svajdler described such a case in 2016 [23]. Given that it is the single reported case of such a pattern of invasion in EC, its prognostic value cannot be assessed. Further studies are necessary to validate this recently described pattern of MI and to provide clinicopathological correlations, mainly regarding the prognostic significance [27].

SCI is characterized by single cells or groups of eosinophilic cells not forming a definite structure, frequently lying in an edematous or myxoid background. SCI is considered as a separate MI pattern by Euscher et al. It is seen in 17% of EECs, most of which also had MELF, although the opposite was not true (the majority of cases with MELF did not have SCI). The authors suggest that there is a possibility that SCI represents an evolving, more aggressive variant of MELF [5]. SCI was previously included within the spectrum of MELF by other authors [18,21,33,34]. Regardless of whether SCI is part of the MELF spectrum, its presence should be regarded as possibly predictive of lymph node metastasis (LNM). Thus, SCI increases the risk of advanced-stage disease. Additional studies are required to determine whether SCI should be considered an additional risk factor when evaluating the need for adjuvant therapy in the unstaged patient [5].

MELF pattern is reported with highly variable frequency ranging between 5.8% and 48% [5,35]. The morphological pattern was initially recognized by Lee et al. in 1994 [36]. Still, the original description was given in 2003 by Murray et al. that settled the acronym MELF as it captures the three main histologic features [21]:1.Microcysts, lined with cells with abundant eosinophilic cytoplasm and vaguely squamoid appearance, or with flattened cells. Lumens often contain neutrophils and occasionally eosinophils.2.Elongated structures lined with the same types of cells and containing the same inflammatory cells in their lumens.3.Clusters of detached cells or individual cells lying in edematous or myxoid tissue.

In cases where the criteria for MELF diagnosis are incomplete, it is helpful to consider this pattern when at least two features are accomplished [37].

Initially, MELF pattern was thought to represent degenerative changes in tumor glands, strongly associated with a stromal reaction ranging from fibromyxoid (the most frequent) to compact and cellular [21]. Later studies suggested that they most frequently represent areas of intense tumor activity similar to EMT observed at the advancing margin of other neoplasms [13,16,34,38,39]. The MELF pattern of MI is deceptive and may be easily overlooked [10,21,40] (Figure 3).

It may be misleading in tumor staging, especially when MELF, often seen focally at the deepest invasive front, is associated with other patterns of MI and, therefore, its recognition is essential to the correct measurement of MI [10,20,21,40,41]. MELF may also be incorrectly interpreted as LVI [10,34,40,42], which is defined as a presence of tumor cells in a space lined by endothelial cells outside the immediate invasive border [42] (Figure 4).

Strong and diffuse expression of CK7 in MELF, SCI, and intravascular tumor cells is helpful for the identification of relatively subtle MELF patterns and otherwise inconspicuous singly infiltrating tumor cells, including those in vessels [21,38].

The possibility that MELF represents a type of EMT becomes increasingly popular [39,43]. EMT was first described by Elizabeth Hay in 1995 [44]. It is a biologic process that allows an epithelial cell to assume a mesenchymal phenotype which includes loss of polarity, acquiring fusiform shape, and production of extracellular matrix components that leads to a change of cell markers; loss of epithelial markers such as E-cadherin, beta-Catenin, EMA and expression of mesenchymal markers vimentin, SMA, fibronectin. A significant hallmark of EMT is the process called “cadherin switch”, that assumes the progressive loss of E-cadherin expression because of intercellular junction disassembly, and its replacement by mesenchymal-type cadherins, such as N-cadherin and cadherin-11 [45]. The functional results of EMT include an enhanced migratory capacity, invasiveness, and resistance of apoptosis [36,45,46]. The phenotypic plasticity afforded by an EMT is revealed by the occurrence of the reverse process—a mesenchymal-epithelial transition (MET) which involves the conversion of mesenchymal cells to epithelial derivatives, but relatively little is known about this process. Importantly, cancer cells may pass through EMT to differing extents, with some cells retaining many epithelial traits while acquiring some mesenchymal ones and other cells shedding all vestiges of their epithelial origin and becoming fully mesenchymal. It is still unclear what specific signals induce EMT in carcinoma cells [46]. Many studies have analyzed different panels of immunohistochemical stains in EEC. Immunohistochemical changes in MELF, in comparison to the parenchyma of EEC, supporting the concept that MELF represents a form of EMT, are as follows:1.Markers of EMT:Markedly decreased E-cadherin expression, a common loss of ER and PR expression [17,21,39,47]. Tumor cells with E-cadherin downregulation involve the process of EMT with an associated increase in invasiveness and metastatic potential and consequently poor prognosis [17,21,48];Lack of mitotic activity and Ki67 shows 0–5% positive neoplastic cells [13,39,49]. This appears to be a paradoxical finding, as invasion and proliferation are generally considered to occur synchronously during tumour progression. But at least in some neoplastic contexts, invasion and proliferation may be related inversely and occur as sequential events, perhaps corresponding to EMT and MET-like phases, respectively. This could also explain why cells exhibiting EMT are less responsive to chemotherapy and radiotherapy, a feature shared with CSCs [43].Reduced galectin-3 expression [50].Aberrant beta-catenin expression (loss or fragmentation of membranous expression) [17,21,37].Stronger fascin expression in tumor cells and stromal cells in MELF. Fascin potentiates the migratory capacity of cells [51].Overexpression of cyclin D1(that usually correlates with LNM and poor prognosis of EEC) [52] and p16 [13,37].

It may be debated whether these alterations necessarily represent an EMT-like process, but they demonstrate localized changes in cellular morphology and immunophenotype at the invasive margin that are presumably transient in nature and likely to be driven by stromal interactions [43].
2.High expression of CSCs markers—CD44, CD133, Nanog1, Sal-like protein 4 (Sall4) [17], S100A4 correlates to MI and LVI [53], VEGF [54].3.Stronger CK7 and CK19 expression that may be helpful in the identification of delicate foci of MELF [37,38,51,55].

The downregulation of hormone receptors expression in EC may be significant for invasion and metastasis and, added to the expression of CSCs markers and loss of E-cadherin expression, lead to the hypothesis that CSCs possess the capability of EMT [56]. The EMT theory, supported by the immunohistochemical evidence helps to explain why the MELF pattern of MI seems to be more vasotropic [20].

MELF pattern is strongly associated with low-grade EEC [5,13,25,34,41,47] and mucinous differentiation in the tumor. The association of focal mucinous differentiation and MELF pattern in EEC cannot be readily explained since the mucinous phenotype within EEC occurs most commonly within the superficial aspects of such tumours in contrast to the typical distribution of MELF along the deep tumour margin. It seems possible that specific genetic alterations might predispose a sub-group of EEC to show both mucinous and MELF features [34].

Many studies have described MELF pattern in association with some poor prognostic factors such as deep MI [38,57], LVI [17,38], LNM [17,33,38], and high-grade tumor [57]. The correlation with LVI and LNM was further confirmed in cases with low-grade EEC [5,13,25,34,35,41]. In this context, MELF pattern in low-grade EEC significantly increases the risk of regional LNM [25,35,41] and could indicate subsequent lymphadenectomy [58]. Quick et al. have also found an association with LVI in low-grade, low-stage EEC but without LNM. Because of the relatively small depth of MI seen in these cases with LVI, they suggest that invasive glandular morphology may be more indicative of LVI risk than the depth of invasion [18]. Hertel et al. noticed that the pattern of LVI and LNM in cases with MELF is more subtle, usually consisting of individual tumor cells or small clusters of cells mimicking histiocytes that may be easily overlooked [41]. Han et al. observed that LVI and MELF patterns of MI are more frequently seen in patients with FIGO grade I EEC with occult LNM. They even proposed an algorithm for identifying occult LNM, including identifying MELF and the careful evaluation of LVI, especially if MELF is present [47].

The prognostic value of the MELF pattern of MI remains controversial. Kihara et al. found no statistically significant association between MELF pattern and recurrence-free survival or disease-free survival in low-grade EEC [14]. Thus, they questioned the casual relation between MELF pattern and LNM, which, in their opinion, can be determined by other associated factors such as tumor size, MI, and LVI. In this view, MELF glands could only represent a morphological change in senescent glands, having no impact on prognosis, as found by Murry et al. [13,21]. Pavlakis et al. also concluded that MELF pattern did not worsen the prognosis of the patients since none recurred or died from the disease. However, they found MELF pattern to be statistically related to LNM [33] and should prompt a particularly careful search for vascular invasion by performing more sections or using immunohistochemistry [10,33]. Some researchers accept that the MELF pattern is significant for a poor prognosis [5,17,38], others regard it as a specialized variant of the infiltrative pattern of invasion, given their frequent concomitancy and the similar molecular changes regarding EMT and stem cell phenotype and may select a group of patients requiring an aggressive therapy [17].

Further studies are needed to evaluate the clinicopathologic implications and prognostic significance of this pattern of MI [33,38]. Due to the proven poor prognosis of MELF pattern in other cancers, some authors recommend that the histopathological report should include information about the presence of MELF pattern [10,27], and clinicians should be aware of MELF’s significance as a more aggressive type of MI to adjust the therapy. They also suggest studying the characteristic molecular expression of type 2 EC in low-grade cancers with aggressive patterns of MI, as these could also represent valuable predictive factors [27].

Recently, The Cancer Genome Atlas (TCGA) project stratified EC in four prognostic groups based on genomic features: an ultramutated phenotype caused by POLE mutations, a hypermutated phenotype caused by DNA mismatch repair deficiency and the resulting microsatellite instability, a copy number-low phenotype and a copy number-high phenotype [59]. A group of Chinese scientists found that POLE mutations in EC combined with histological characteristics, including high stage, deep MI, and especially MELF pattern, predicted poor progression-free survival. Thus, integrating POLE mutation status with established clinicopathologic factors, including stage, MI and MELF, in the risk assessment of EC is more effective and might lead to a precise and individualized therapeutic strategy [60].

No statistically significant molecular aberrations were found to be more frequently associated with MELF in comparison to other conventional patterns of MI [61]. Considering the possibility to identify specific molecular markers consistent with these patterns of MI, in correlation with EMT and CSCs, future targeted therapies open promising perspectives in the treatment of EEC [27].

## 4. Conclusions

The assessment of MI depth and correct staging of EC is possible only after precise recognition of each MI pattern. More aggressive patterns of MI are diffusely infiltrative pattern and MELF pattern, but little is known about SCI, adenoma malignum-like pattern, and phyllodes tumor-like pattern. Deceptive patterns of MI, which may be easily overlooked, include MELF, SCI, and adenoma malignum-like pattern. Despite the still controversial prognostic value of MELF, we agree that it has to be actively searched for and assessed together with LVI by pathologists. Both phenomena should be recorded in histopathological reports to make surgeons aware of a worse influence on prognosis.

## Figures and Tables

**Figure 1 diagnostics-11-01707-f001:**
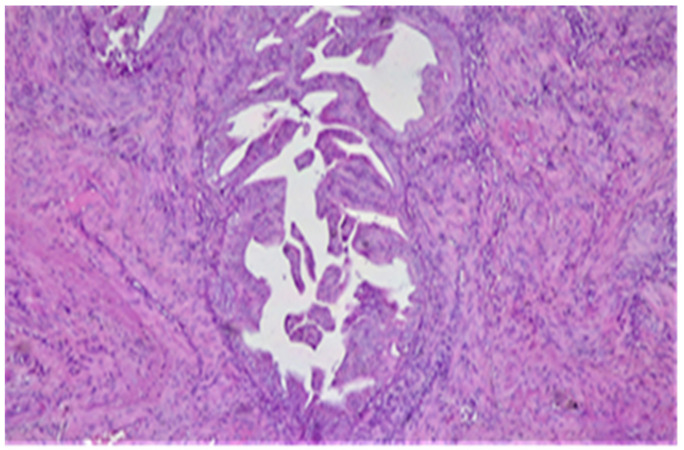
Adenomyosis-like type of invasion—groups of malignant glands invading the myometrium without any surrounding stroma. HE × 10.

**Figure 2 diagnostics-11-01707-f002:**
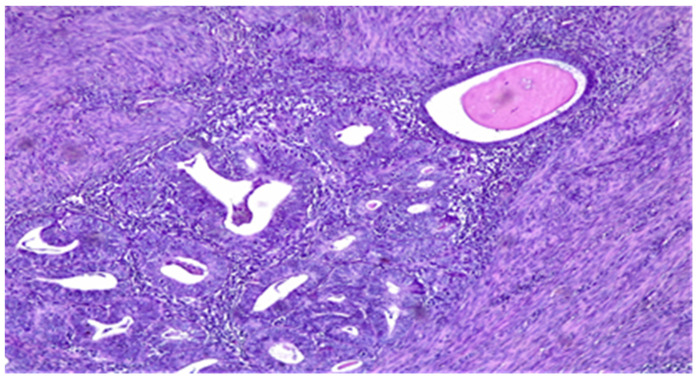
Adenomyosis involved by EEC—residual endometrial stroma and a single normal gland (top right) around the tumor nest. HE × 10.

**Figure 3 diagnostics-11-01707-f003:**
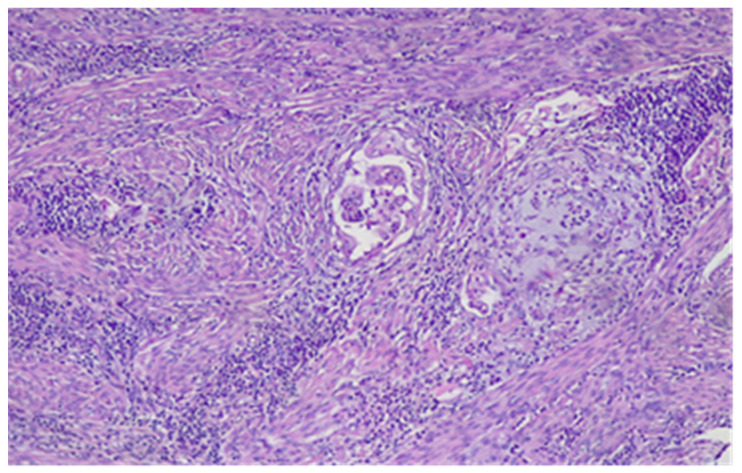
MELF pattern of MI. HE × 10.

**Figure 4 diagnostics-11-01707-f004:**
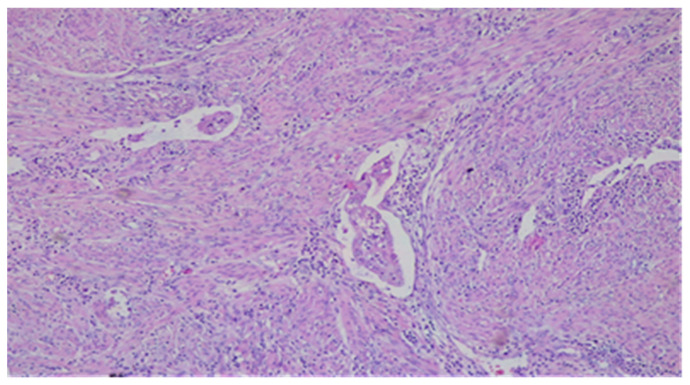
LVI—a single vessel with tumor embolus in the lumen. HE × 10.

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
