# Peer review of "Patterns of Myometrial Invasion in Endometrial Adenocarcinoma with Emphasizing on Microcystic, Elongated and Fragmented (MELF) Glands Pattern: A Narrative Review of the Literature"

_diagnostics, 2021, doi:10.3390/diagnostics11091707_

Round 1

Reviewer 1 Report

This manuscript is a review of the literature on patterns of myometrial invasion in endometrial cancer. It is not a quantitative or qualitative study. 

Major Comment

  1. What is more important - the pattern or the Immunohistochemical staining, genetics (MMR proficient or not), issues like POLE ect?
  2. With so many histologic patterns again how likely is it that the kappa values for pattern recognition among pathologists would be high (ie., agree)?

3. How do the authors suggest moving forward formal research into the meaning of MELF?

Minor comments

The manuscript becomes easy to read after the first 1.5pages. Those first pages could use editing of the english grammar, word choice and phrasing of sentences. For example line 32 - is it estrogen background or estrogen influence. Lines 30-34 could be more clearly stated. 

Angel Yordanov is spelled angel.jordanov@gmail.com. Check if the last name is with a J or a Y.

EMT appears on page 5 but it is never defined in the verbage. 

Line 295 - Suggest to delete the word "conclusion"

Consider a table to show the conflicting studies about MELF.

Author Response

This manuscript is a review of the literature on patterns of myometrial invasion in endometrial cancer. It is not a quantitative or qualitative study.

Author’s reply: Thank you for this review.

Major Comment

  1. What is more important - the pattern or the Immunohistochemical staining, genetics (MMR proficient or not), issues like POLE ect?

Author’s reply: The purpose of this review is to present the latest and most important information on the morphological patterns of myometrial invasion in endometrial cancer and its role in proper staging. We have tried to summarize the available information published so far for each individual model, focusing in more aggressive types of myometrial invasion and mainly on MELF. We discuss the morphology, the diagnostic challenges they pose to pathologists, the immunohistochemical characteristics and the results of various studies on their prognostic value. Our review does not aim to prioritize the different methods for assessing endometrial cancer - morphological, immunohistochemical, genetic, molecular or others. Each of them has its significance and disadvantages, which are not the subject of this manuscript. There is currently no universal scientific method that gives an accurate prognostic assessment of endometrial cancer and therefore continue research in various fields / morphological, immunohistochemical, genetic, etc./.

  1. With so many histologic patterns again how likely is it that the kappa values for pattern recognition among pathologists would be high (ie., agree)?

Author’s reply: The patterns of myometrial invasion are really many and are apparently increasing. This is one of the reasons to pay detailed attention to them, as well as the fact that they vary in number and overlap in different studies. We think that the kappa values ​​for pattern recognition among pathologists would not be high. The accumulation of more information about the influence of myometrial invasion models on the development of endometrial cancer will most likely lead to their aggregation and selection of a specific number of prognostically significant morphological features. For now, it is necessary to know and distinguish the whole spectrum of models of myometrial invasion.

  1. How do the authors suggest moving forward formal research into the meaning of MELF?

Author’s reply: We suggest searching for specific molecular markers consistent with MELF and further studies for evaluating prognostic significance of this pattern of myoinvasion

Minor comments

The manuscript becomes easy to read after the first 1.5pages. Those first pages could use editing of the english grammar, word choice and phrasing of sentences. For example line 32 - is it estrogen background or estrogen influence. Lines 30-34 could be more clearly stated.

Author’s reply: : Thank you. The whole manuscript was revised by native speaker.

Angel Yordanov is spelled angel.jordanov@gmail.com. Check if the last name is with a J or a Y.

Author’s reply: Thank you. It is spelled correct.

EMT appears on page 5 but it is never defined in the verbage.

Author’s reply: Thank you. It is fit as recommended.

Line 295 - Suggest to delete the word "conclusion"

Author’s reply: Thank you. It is done as recommended.

Consider a table to show the conflicting studies about MELF.

Author’s reply: We discussed a possible table, but the cited studies do not compare the same categories of endometrial carcinomas / low grade, low stage, high grade /, do not use the same statistical methods for analysis and give different interpretations of the results found. Therefore, we decided to cite the results of these studies and the way in which their authors interpret them, as such information could not be provided in tabular form.

Reviewer 2 Report

This is a well written review presenting the most recent information about more aggressive patterns of MI, in particular MELF.

Considering myometrial invasion and The Cancer Genome ATLAS (TCGA) groups as crucial independent risk factors in EC, please add some data from available studies (e.g.doi: 10.1016/j.ygyno.2020.07.102; DOI: 10.1016/j.ygyno.2021.05.029).
